# Reduced Cholesterol Levels during Acute Human Babesiosis

**DOI:** 10.3390/pathogens12040613

**Published:** 2023-04-18

**Authors:** Luis A. Marcos, Charles Kyriakos Vorkas, Inderjit Mann, Evan Garry, Pooja Lamba, Sophia K. Pham, Rachel Spector, Aikaterini Papamanoli, Sara Krivacsy, Michael Lum, Aleena Zahra, Wei Hou, Eric D. Spitzer

**Affiliations:** 1Division of Infectious Diseases, Department of Internal Medicine, Stony Brook University, Stony Brook, NY 11794, USA; 2Department of Microbiology and Immunology, Stony Brook University, Stony Brook, NY 11794, USA; 3Stony Brook Southampton Hospital, Southampton, NY 11968, USA; 4Division of Epidemiology and Biostatistics, Department of Family, Population and Preventive Medicine, Stony Brook University, Stony Brook, NY 11794, USA; 5Department of Pathology, Stony Brook University, Stony Brook, NY 11794, USA

**Keywords:** *Babesia*, lipids, high-density lipoprotein, low-density lipoprotein, cholesterol

## Abstract

Background: Babesiosis, an intra-erythrocytic protozoan disease, is an emerging zoonotic parasitic disease worldwide. Cholesterol levels are correlated with severe infections, such as sepsis and COVID-19, and anecdotal reports suggest that high-density lipoprotein (HDL) cholesterol declines during acute babesiosis. Our aim was to describe the cholesterol levels in patients with acute babesiosis diagnosed in an endemic area in New York, hypothesizing that HDL levels correlate with the severity of infection. Methods: We reviewed the medical records of adult patients with babesiosis diagnosed by identification of *Babesia* parasites on a thin blood smear and confirmed by polymerase chain reaction from 2013 to 2018, who also had available a lipid profile drawn at the time of clinical presentation. Additional lipid profile levels were considered as “baseline” if they were drawn within 2 months before or after the infection as part of routine care. Results: A total of 39 patients with babesiosis had a lipid profile drawn on presentation. The patients were divided into two groups for comparison based on the treating physician’s clinical decision: 33 patients who were admitted to the hospital and 8 patients who were evaluated as outpatients. A history of hypertension was more common in admitted patients (37% vs. 17%, *p* = 0.02). The median levels of low-density lipoprotein (LDL) and HDL were significantly reduced in admitted patients compared to non-admitted patients (46 vs. 76 mg/dL, *p* = 0.04; and 9 vs. 28.5 mg/dL, *p* = 0.03, respectively). In addition, LDL and HDL levels returned to baseline values following resolution of acute babesiosis. Conclusion: LDL and HDL levels are significantly reduced during acute babesiosis, suggesting that cholesterol depletion may predict disease severity. Pathogen and host factors may contribute to a reduction in serum cholesterol levels during acute babesiosis.

## 1. Introduction

Babesiosis, an emergent tick-borne disease caused mainly by *Babesia microti* in the US, is primarily transmitted by the bite of Ixodes deer ticks, and less frequently through blood transfusion, organ transplantation and perinatally [1]. The initial clinical presentation often includes fevers and fatigue but can range from asymptomatic infection to severe disease with end-organ damage. About one-third of hospitalized patients with acute *Babesia* infection require admission to the intensive care unit due to severe anemia, shock, severe parasitemia, renal failure or respiratory distress [2]. Importantly, a subset of patients develop a persistent or relapsing infection that is more common in immunocompromised hosts who have a higher probability of complications [3,4,5].

*Babesia*-associated mortality in the United States has been reported to be 1.6% and has been correlated with a high degree of parasitemia [6]. In severe disease, the use of adjunctive red blood cell exchange is the main therapeutic option to reduce parasitemia, though additional laboratory biomarkers to measure risk for progression to severe disease or clinical improvement are lacking [7].

Common laboratory findings in acute *Babesia* infection include evidence of hemolytic anemia based on decreases in red blood cell count, hemoglobin and haptoglobin, coupled with elevated LDH and indirect bilirubin; however, these biomarkers do not predict mortality [7]. To date, most studies investigating clinical biomarkers for disease severity have focused on the degree of hemolysis and tissue hypoxia and do not reliably correlate with clinical presentation [8]. As babesiosis continues to emerge in a broader geographic distribution [1], with an increase in hospitalized cases [6], particularly in immunocompromised patients with persistent or relapsing disease, or even death [9], the study of novel biomarkers to monitor disease and predict outcomes is a critical priority.

Dysregulation of lipid metabolism has been widely reported during bacterial [9], viral [10] and parasitic infections [11,12]. High-density lipoprotein (HDL) cholesterol has been shown to be markedly reduced in a small case series of acute babesiosis [13,14], although the utility of cholesterol monitoring as a biomarker for predicting disease severity remains unknown. Furthermore, in animal models of babesiosis, changes in lipid profiles have been observed during acute infection [15]. In a dog model using *Babesia canis* infection, the HDL concentration was significantly lower than controls [16]. In vitro studies have shown that HDL is the major source of lipids for the growth of *Babesia divergens* in human erythrocytes [17]. Lipids seem to play an important role during acute infection by *Babesia* in humans and animals, and lower levels of HDL may be a biomarker of severe disease. It is also not known whether low levels of HDL prior to infection affect the severity of acute babesiosis.

The aim of this study is to investigate the correlation of cholesterol levels with disease progression during acute babesiosis with the hypothesis that the lipid profile is markedly dysregulated during severe acute infection.

## 2. Material and Methods

Study design: We conducted a retrospective chart review of adult patients who had a positive peripheral blood thin smear for intra-erythrocyte *Babesia* parasites at Stony Brook University and Southampton Hospitals (Stony Brook Medicine) between 2013 and 2018. Cases were stratified by admission status after initial emergency department evaluation, representing a clinical impression of disease severity requiring hospitalization.

Patient selection: All cases were confirmed to be positive for *B. microti* by PCR analysis performed at the NY State Department of Health. Only patients who had a lipid profile drawn during acute presentation were included in the analysis. Median values of peak parasitemia were defined as the largest percentage of parasites detected on a peripheral blood smear throughout the hospitalization.

Data collection: Manual chart reviews were performed to systematically abstract data for variables including demographics and comorbidities such as hypertension, splenectomy, cardiovascular disease, diabetes, autoimmune diseases, chronic obstructive pulmonary disease and chronic kidney disease. Laboratory variables included parasitemia peak, lipid profile during acute presentation, lipid profile within 2 months before or after the infection (if available in medical records as part of routine clinical care) and hemoglobin.

Longitudinal analysis of lipid profiles: Lipid profile included high-density lipoprotein (HDL), low-density lipoprotein (LDL), very low-density lipoprotein (VLDL) and triglycerides (TG), which were drawn routinely according to the clinician’s criteria during hospitalization of these patients.

Baseline lipid profiles: In order to compare these lipid profiles during acute infection with baseline lipid profiles, a search in the medical record on each case was performed. Historic lipid profile results (within 2 months before or after the infection) on the same patients were collected to obtain a baseline lipid profile when they were not infected (annual physical outpatient visit or routine blood draw).

Statistical analysis: Frequencies and percentages for categorical variables were calculated using a univariate descriptive analysis. As the aim of the study was to assess the role of lipids on the severity of disease, we separated the cases into admitted vs. non-admitted to the hospital as a surrogate criterion for clinical purposes. Comparisons between admitted and non-admitted were performed with categorical variables by Chi-square or Fisher’s exact test and to continuous variables by a Mann–Whitney U test. Pre-post changes in lipid profiles were tested using a Wilcoxon signed rank test. A value of *p* < 0.05 was considered as the critical level of significance. All analyses were performed using SAS v9.4 (the SAS Institute, Cary, NC, USA).

## 3. Results

Study population. A total of 39 patients (29.7% female) met the criteria for acute *Babesia* infection and had a lipid profile drawn on admission (% drawn lipid profile). The median age was 68 (range 19–89). Comorbidities included hypertension (*n* = 19, 54.3%), splenectomy (*n* = 1, 2.8%), cardiovascular disease (*n* = 11, 34.4%), diabetes (*n* = 11, 32.3%), chronic obstructive pulmonary disease (*n* = 7, 20%) and chronic kidney disease (*n* = 4, 11.4%).

Hospitalization. The patients were stratified by admission status after initial emergency department evaluation: 33 patients (median age 67 years-old; range: 56–83.5) who were admitted and 8 patients (median age: 72.5 years-old; range 67–80) who were discharged with outpatient follow-up (*p* = 0.5). Median LDL and HDL were significantly reduced in admitted patients compared to non-admitted patients (46 vs. 76 mg/dL, *p* = 0.04; and 9 vs. 28.5 mg/dL, *p* = 0.03, respectively) (Table 1).

Lipid profile analysis. To further investigate the lipid profile in these cases, and after comparing them with baseline cholesterol levels (approximately 2 weeks to 2 months apart from acute infection), we found significant changes in all cholesterol levels when compared to these levels during and pre- and post-infection (Table 2) and in admitted versus non-admitted patients (Figure 1). A correlation analysis between parasitemia and cholesterol levels (before, during and after the infection) found no significant association between these two variables (data not shown).

## 4. Discussion

In patients with acute babesiosis who were admitted to hospital, we found that LDL and HDL levels were significantly reduced compared to milder cases who were discharged home after initial ED assessment. Patients with low HDL levels during acute *Babesia* infection also had baseline HDL levels within normal limits in the 2 months before and/or after the infection, which strongly suggests that preexisting HDL deficiency (e.g., due to lecithin-cholesterol acyltransferase deficiency) did not predispose these patients to acute babesiosis. Undetectable HDL has been also found in malaria, a similar protozoan infection [18]. The mechanism of this reversible low HDL phenomenon in *Babesia* infections is unknown; however, several lines of evidence suggest that it may be indirectly related to the host inflammatory response and/or directly related to the growth of the *Babesia* organisms.

In bacterial sepsis, low HDL levels inversely correlate with the severity of septic disease and are associated with an exaggerated systemic inflammatory response [19]. HDLs contain two main proteins, apolipoprotein A-I (apoA-I) and apoA-II. Interactions between parasites and HDL may be related to specific apolipoproteins. For example, apoL-1 has been associated with innate immunity properties against other protozoan infections including *Trypanosoma brucei* and *Leishmania* spp. [20,21].

Data from the malaria literature suggest that decreases in HDL associated with babesiosis may be directly related to the growth of the parasite. Multiple studies have shown that *Plasmodium* spp., Apicomplexans protozoa that are closely related to *Babesia* spp., does not possess the biosynthetic pathway for de novo synthesis of cholesterol and must scavenge this important molecule from host hepatocyte and erythrocyte membranes [22,23,24]. In humans, acute malaria infections are associated with lower LDL and HDL and the total cholesterol [25]. Another study showed that *P. falciparum*-infected individuals had significantly lower cholesterol levels than controls [26].

The limitations of the study include that not all patients had a lipid profile on admission, as this is not a routine test for patients evaluated for acute babesiosis.

In conclusion, HDL and LDL levels are markedly reduced in acute human babesiosis, warranting hospitalization. Whether HDL and LDL levels may correlate with clinical outcomes, including relapsing disease, or may be useful in monitoring patient response to therapy will be the subject of future investigations. Further studies are needed to elucidate the mechanisms underlying the depletion of HDL and LDL during acute human babesiosis.

## Figures and Tables

**Figure 1 pathogens-12-00613-f001:**
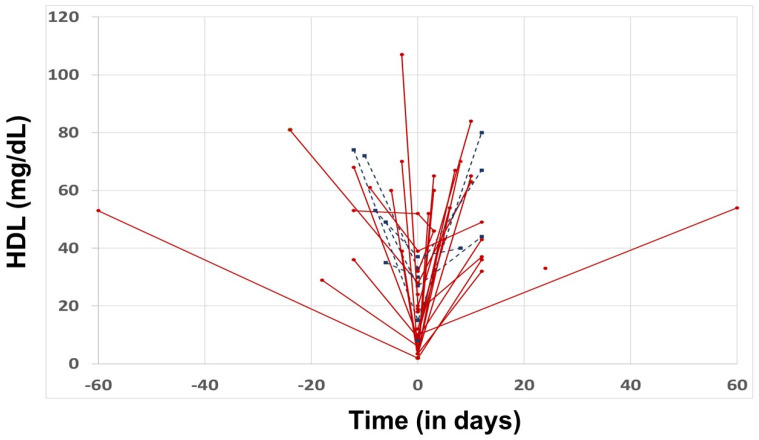
HDL levels in acute babesiosis (*x* axis: left of zero = days prior to diagnosis of acute babesiosis; right of zero = levels drawn post-acute infection; zero = day 0 of acute infection; *y* axis: level of HDL in mg/dL). Blue dots represent patients who were not admitted to hospital. Red dots represent patients who were admitted. Note: HDL levels markedly decreased during acute infection, returning to baseline levels after infection during follow up.

**Table 1 pathogens-12-00613-t001:** Numeric variables compared between admitted versus non-admitted patients with acute babesiosis.

	Admitted		Not Admitted		
Variable	N	Median (IQR)	N	Median (IQR)	*p* Value
Age	33	67.00 (56.00–83.50)	6	72.50 (67.00–80.00)	0.57
Parasitemia (%)	33	0.90 (0.30–2.65)	6	0.40 (0.10–0.50)	0.08
Hemoglobin (g/dL)	33	11.60 (9.85–13.00)	6	12.65 (12.00–13.20)	0.30
Total Cholesterol	31	104.00 (84.00–129.00)	6	119.00 (106.00–127.00)	0.29
LDL (mg/dL)	23	46.00 (32.00–67.00)	5	76.00 (59.00–85.00)	0.04
HDL (mg/dL)	31	9.00 (3.50–19.00)	6	28.50 (15.00–33.00)	0.03
TG (mg/dL)	31	190.00 (123.00–311.00)	6	110.00 (100.00–121.00)	0.07
VLDL (mg/dL)	29	38.00 (25.00–55.00)	6	22.00 (20.00–24.00)	0.09

LDL: Low-density lipoprotein. HDL: High-density lipoprotein. TG: triglycerides, VLDL: very-low-density lipoprotein. All lipid profiles were drawn within 24 h of acute presentation in the emergency room.

**Table 2 pathogens-12-00613-t002:** Change in cholesterol levels during *Babesia* infection compared to “pre” and “post” infection (more than 1 month before or after the infection).

Variable	Change	N	Median	Lower Quartile	Upper Quartile	*p* Value
Total Cholesterol	[acute]-[pre]	16	−73	−95.5	−33	0.0002
	[post]-[acute]	23	61	25	108	<0.0001
LDL	[acute]-[pre]	12	−26.5	−54.5	3.5	0.082
	[post]-[acute]	15	37	26	55	0.0006
HDL	[acute]-[pre]	16	−37.25	−58.5	−22	<0.0001
	[post]-[acute]	23	35	19	58	<0.0001
TG	[acute]-[pre]	16	58.5	15.5	175.5	0.0013
	[post]-[acute]	23	−72	−168	−24	<0.0001
VLDL	[acute]-[pre]	15	10	0	32	0.0022
	[post]-[acute]	21	−14	−26	−5	0.0002

## Data Availability

Data is contained within the article.

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
