# Peer review of "Reduced Cholesterol Levels during Acute Human Babesiosis"

_pathogens, 2023, doi:10.3390/pathogens12040613_

Round 1

Reviewer 1 Report

Marcos and coworkers present a simple study documenting HDL and LDL levels in Babesia species infected adult patients recruited at the Stony Brook University hospital in New York from 2013 to 2018. They have investigated patients at the acute stage of Babesia infection which was confirmed by parasitemia determination by microscopic examination of thin blood smears as well as by qPCR. It is a clearly written short article. The major problem is presentation of data in the figure 1. It is difficult to visualize individual patient information in this figure. I think a bar diagram using different colors for hospitalized and non-hospitalized patients with individual patient code number marked in the middle of each bar (it can simply be designated by serial numbers) will provide information to readers in a more understandable manner. In addition, the font size for the days of infection in the figure is too small and blurry with respect to the size of figure and the font used to explain days under the figure. A proportionate size font should be used in different parts of this figure.

Minor points:

1.     Babesiosis should be written with small font when in the middle of the sentence such as in line 55. Babesia should be italicized throughout, not just in line 166.

2.     Second sentence in Table 2 heading is not needed.

3.     Figure 1 legend: HDL levels at acute babesiosis state---- may be better.

4.     Spp is usually not italicized (for example line 156).

Author Response

Reviewer 1

Marcos and coworkers present a simple study documenting HDL and LDL levels in Babesia species infected adult patients recruited at the Stony Brook University hospital in New York from 2013 to 2018. They have investigated patients at the acute stage of Babesia infection which was confirmed by parasitemia determination by microscopic examination of thin blood smears as well as by qPCR. It is a clearly written short article. The major problem is presentation of data in the figure 1. It is difficult to visualize individual patient information in this figure. I think a bar diagram using different colors for hospitalized and non-hospitalized patients with individual patient code number marked in the middle of each bar (it can simply be designated by serial numbers) will provide information to readers in a more understandable manner. In addition, the font size for the days of infection in the figure is too small and blurry with respect to the size of figure and the font used to explain days under the figure. A proportionate size font should be used in different parts of this figure.

Response: We would like to keep Figure 1 in its present configuration since it provides both temporal data as well as the changes in HDL levels. It would be difficult to show the rate of decline and recovery of HDL levels using bar graphs. We agree that the font is difficult to read on both axes and have adjusted this to make the values more legible.

Minor points:

  1. Babesiosis should be written with small font when in the middle of the sentence such as in line 55. Babesia should be italicized throughout, not just in line 166.

Response: All Babesia words were italicized.

  1. Second sentence in Table 2 heading is not needed.

Response: Second sentence in table 2 has been deleted.

  1. Figure 1 legend: HDL levels at acute babesiosis state---- may be better.

Response: Done.

  1. Spp is usually not italicized (for example line 156).

Response: All spp has been changed and it is no longer italicized.

Reviewer 2 Report

COMMENTS

The manuscript presented describes the cholesterol levels in patients with acute babesiosis. The manuscript is quite well and clearly written. However, the authors must include in the introduction the novelty of their study as the alterations of lipid profile especially the reduction of HDL and LDL cholesterol in acute babesiosis are already reported in both human and animals (dogs/cattle). Moreover, serum biomarkers for clinical monitoring of babesiosis including the main protein component of HDL, apolipoprotein A-I (apoA-I) have been identified and reported in canines.

Reference 12 is not cited in the text which may be included.

Material and methods:

Patient selection: It is not very clear whether the patients selected for the study with parasitemia had any reported comorbidities or only the patients exclusively with parasitemia were selected for the study

Data collection: “Laboratory variables included parasitemia peak, lipid profile during acute presentation, lipid profile within 2 months before or after the infection, and hemoglobin.”

It may be clarified how lipid profile data was collected 2 months before the infection.

Author Response

Reviewer 2

The manuscript presented describes the cholesterol levels in patients with acute babesiosis. The manuscript is quite well and clearly written. However, the authors must include in the introduction the novelty of their study as the alterations of lipid profile especially the reduction of HDL and LDL cholesterol in acute babesiosis are already reported in both human and animals (dogs/cattle). Moreover, serum biomarkers for clinical monitoring of babesiosis including the main protein component of HDL, apolipoprotein A-I (apoA-I) have been identified and reported in canines.

Response: In addition to following sentence “High-density lipoprotein (HDL) cholesterol has been shown to be markedly reduced in a small case series of acute babesiosis [13,14], although the utility of cholesterol monitoring as a biomarker for predicting disease severity remains unknown”, we added the following paragraph in the introduction: “Furthermore, in animal models of babesiosis, changes in lipid profile were observed during acute infection [15]. In a dog model using Babesia canis infection, the HDL concentration was significantly lower than controls [16]. In vitro studies have shown that HDL was the major source of lipids for the growth of Babesia divergens in human erythrocytes [17]. Lipids seem to play an important role during acute infection by Babesia in humans and animals, and lower levels of HDL may be a biomarker of severe disease”. It is also not known whether low levels of HDL prior to infection affect the severity of acute babesiosis.

Reference 12 is not cited in the text which may be included.

Response: Ref 12 was added. 

Material and methods:

Patient selection: It is not very clear whether the patients selected for the study with parasitemia had any reported comorbidities or only the patients exclusively with parasitemia were selected for the study.

Response: Patients were selected only because of evidence of parasitemia. Some of them have comorbidities. Line 105: “Comorbidities included hypertension (n=19, 54.3%), splenectomy (n=1, 2.8%), cardiovascular disease (n=11, 34.4%), diabetes (n=11, 32.3%), chronic obstructive pulmonary disease (n=7, 20%) and chronic kidney disease (n=4, 11.4%)”.

Data collection: “Laboratory variables included parasitemia peak, lipid profile during acute presentation, lipid profile within 2 months before or after the infection, and hemoglobin.”

It may be clarified how lipid profile data was collected 2 months before the infection.

Response: We added “…infection (if available in medical records as part of routine clinical care), and hemoglobin”. We only captured the lipid profile on patients who had it performed at the discretion of their primary care physician. This was also explained in the original paper on line 88:

“Baseline lipid profiles: In order to compare these lipid profiles during acute infection with baseline lipid profiles, a search in the medical record on each case was performed. Historic lipid profiles results (within 2 months before or after the infection) on the same patients were collected to obtain a baseline lipid profile when they were not infected (annual physical outpatient visit or routine blood draw”.